# The metabolic ability of swallowtails results in the production of bioactive substances from plant components

Miho Nakano[1,2¤a], Takuma Sakamoto[2], Yoshiyuki Itoh[3], Yoshikazu Kitano[4], Kaori Tsukakoshi[5¤b], Hidemasa Bono[6,7], Hiroko Tabunoki[1,2*]

1 Cooperative Major in Advanced Health Science, Graduate School of Bio-Applications and System Engineering, Tokyo University of Agriculture and Technology, Tokyo, Japan, 2 Department of Science of Biological Production, Graduate School of Agriculture, Tokyo University of Agriculture and Technology, Tokyo, Japan, 3 Smart-Core-Facility Promotion Organization, Tokyo University of Agriculture and Technology, Tokyo, Japan, 4 Department of Applied Biological Science, Tokyo University of Agriculture and Technology, Tokyo, Japan, 5 Department of Biotechnology and Life Science, Tokyo University of Agriculture and Technology, Tokyo, Japan, 6 Laboratory of Bio-DX, Genome Editing Innovation Center, Hiroshima University, Hiroshima, Japan, 7 Laboratory of Genome Informatics, Graduate School of Integrated Sciences for Life, Hiroshima University, Hiroshima, Japan

¤a Current address: Department of Food & Cosmetic Science, Faculty of Bioindustry, Tokyo University of Agriculture, Hokkaido, Japan
¤b Current address: Department of Chemistry, Faculty of Science, Tokyo University of Science, Tokyo, Japan
* h_tabuno@cc.tuat.ac.jp

## Abstract

Host plant selection may depend on the metabolic system in herbivorous insects. Although oligophagous insects take up specific host plant components, how host plant components and their biological activities are altered through their metabolic systems remains unknown. Here, by examining gene expression of metabolic enzymes and components in the larval frass, we investigated the metabolic ability of *Papilio memnon* larvae fed with *Citrus* x *paradisi* (grapefruit) against host plant components. The gene expression levels of some metabolic enzymes were fluctuated between the larval midgut and the larval fat bodies. Furthermore, the chloroform extract from the larval frass, but not that from grapefruit leaves, inhibited cell viability of human pancreatic cancer cell line, MIA PaCa2. Finally, we identified two chlorophyll catabolites, pheophorbide-a and pyropheophorbide-a, in the larval frass extract. Pyropheophorbide-a reduces cell viability of and induces morphological changes in cells of MIA PaCa2; in addition, pheophorbide-a and pyropheophorbide-a inhibit the aggregation of amyloid β-protein (human, 1–42). Therefore, the chemical structure and biological activity of host plant components are affected by the *P. memnon* metabolic system. Our findings may contribute to the understanding of the process for producing pheophorbide-a and pyrophephorbide-a from chlorophyll, facilitated by the metabolic ability of *P. memnon* larvae.

**Data availability statement:** All relevant data are within the paper and its Supporting Information files. The RNA sequencing datasets generated and/or analysed during the current study are available in the Sequence Read Archive, DNA Data Bank of Japan repository, under the following accession IDs: fat body groups (DRR619748, DRR619750, and DRR619752) and midgut groups (DRR619749, DRR619751, and DRR619753).

**Funding:** This work was supported by JSPS KAKENHI grant numbers 21J21584 to MN, JSPS KAKENHI grants 22K14899 to TS, and JSPS KAKENHI grants 20K20571 and 23K17418 to HT. This work was also supported by the Center of Innovation for Bio-Digital Transformation (BioDX), an open innovation platform for industry–academia cocreation (COI-NEXT), Japan Science and Technology Agency (JST, COI-NEXT,JPMJPF2010), provided to H.B. and H.T. The ROIS-DS-JOINT grant 004RP2017 and 016RP2018 were provided to H.B. and H.T.

**Competing interests:** The authors have declared that no competing interests exist.

**Abbreviations:** Phe-a, Pheophorbide-a; Pph-a, Pyropheophorbide-a; Chl-a, Chlorophyll-a; DMSO, Dimethyl sulfoxide; EGCG, Epigallocatechin gallate; ThT, Thioflavin T; HFIP, Hexafluoroisopropanol; Aβ, Amyloid β-protein; CYP, Cytochrome P450; GST, Glutathione S-transferases; CCE, Carboxylesterase; EH, Epoxide hydrolase; TLC, Thin-layer chromatography; PS, Photosensitizer; PDT, Photodynamic therapy; RFP, Red fluorescent protein; CHBP, Chlorophyllide binding protein.

## Introduction

The frass of some herbivorous insects is utilized as a natural medicine because it is believed to be effective in the treatment of symptoms such as inflammation [1]. For example, frass from a stick insect, *Eurycnema versifasciata*, is used to treat diarrhea, asthma, upset stomach, and muscular pain in Malaysia [2]. In China, frass from *Bombyx mori* (silkworm) larvae living on mulberry leaves is thought to stimulate the adrenal gland and decrease the blood cholesterol content [3]. Furthermore, some biological activities of the extract from *B. mori* frass have been reported. The ethanol extract has potential antiallergic activities because it decreases the production of several Th2-related cytokines, thus alleviating food allergy symptoms and allergic contact dermatitis [4,5]. The acetone extract decreased damage to neurons caused by amyloid-β oligomers, and it reversed amyloid-β oligomer-induced memory loss in mice [6]. The methanol extract inhibited the proliferation of the human colon cancer cell line HT-29 and induced caspase-mediated apoptosis [7].

Swallowtails feed on specific plants containing bioactive compounds during the larval phase.

The swallowtails *Papilio xuthus* (Asian swallowtail) and *Papilio machaon* (yellow swallowtail) feed on Rutaceae plants and Apiaceae plants, respectively, which contain many pharmaceutical compounds. Therefore, these plants have been used as natural medicines [8–12]. Our previous study revealed that a chloroform extract from *Citrus trifoliata*-fed *P. xuhus* larval frass reduced the viability of the human pancreatic cancer cell line MIA PaCa2 [13]; moreover, a chloroform extract from *Angelica keiskei*-fed *P. machaon* larval frass reduced the viability of the human colon cancer cell line HCT116 [14]. The activities of the frass extracts were stronger than those of the host plant. Accordingly, we speculated that the chemical structure of host plant components might be altered via larval metabolism, leading to a reduction in the viability of human cancer cell lines. The bioactive substances from food plants may be concentrated and excreted via the frass; however, how the host plant components are altered to bioactive substances included in swallowtail larval frass has not been identified because the metabolic pathway has yet to be examined.

*P. memnon* (the great Mormon) and *P. xuthus* feed on Rutaceae plants as host plants. *P. memnon* larvae preferably feed on *Citrus* plants, and *P. xuthus* larvae feed on not only *Citrus* but also *Zanthoxylum*, *Skimmia*, and *Tetradium* plants [15]. These host plants contain furanocoumarins, and host plant selection may be based on the kinds of furanocoumarins produced by the host plants, likely leading to differences in metabolic ability between *P. memnon* and *P. xuthus* larvae. Grapefruit (*Citrus × paradisi*) contains furanocoumarins and flavonoids that have a variety of biological activities, such as anticancer, antioxidative, and anti-inflammatory activities [16,17].

In this study, we investigated metabolic capacity by analyzing the final metabolites in larval frass and transcript levels of metabolic enzymes in the larval midgut and fat bodies of *P. memnon* larvae fed *C. paradisi* leaves.

## Materials and methods

### Insect culture and sample collection

*P. memnon* larvae were caught at the Fuchu campus of Tokyo University of Agriculture and Technology. The larvae were reared on the leaves of *C. paradisi* collected at the Fuchu campus and maintained at 25°C under a 16-hour light/8-hour dark cycle in the laboratory. The larval frass and the host plant leaves were collected and stored at −30°C until use.

### RNA extraction

We purified total RNA from the midgut and fat body of *P. memnon* 5th instar larvae reared on *C. paradisi* using TRIZOL reagent (Thermo Fisher Scientific, Waltham, MA, USA) and a PureLink® RNA Extraction Kit (Thermo Fisher Scientific) according to the manufacturer's protocol.

### RNA sequencing

An Agilent TapeStation 2200 (Agilent Technologies, Santana Clara, CA, USA) was used to evaluate the quality of the RNA from the midgut and fat bodies of *P. memnon* larvae fed *C. paradisi*. cDNA library construction from total RNA extracted from these samples was performed using the NEBNext® Poly(A) mRNA Magnetic Isolation Module and NEB NEXT Directional Ultra RNA Library Prep Kit for Illumina® (New England Biolabs, Waltham, MA, USA) following the manufacturer's instructions. Finally, the libraries (150 bp, paired-end) were sequenced with the Illumina NovaSeq 6000 platform (Illumina Inc., San Diego, CA, USA).

### RNA sequencing data analysis

FASTQ files of RNA sequencing data were assessed via TrimGalore! version 0.6.7 (https://www.bioinformatics.babraham.ac.uk/projects/trim_galore/, accessed on 18 February 2025). The trimmed data were subsequently aligned to the reference genome of nonmimetic *P. memnon* version 1.0 obtained from the NCBI database using HISAT2 version 2.2.1 (http://daehwankimlab.github.io/hisat2/, accessed on 17 February 2025). Next, the aligned data were sorted by using SAMtools version 1.17 (http://www.htslib.org, accessed on 17 February 2025). To estimate expression, StringTie version 2.1.7 [18] was used for the following operation: first, expression was estimated for each set of sorted data. Second, the estimated data were merged. Finally, expression was estimated from each set of sorted data using the merged data as annotations to construct files for Ballgown. To analyze DEGs between the fat bodies and midgut by edgeR [19], read count data of transcripts were obtained from files for Ballgown via the prepDE Python script provided by the authors of StringTie (https://ccb.jhu.edu/software/stringtie/dl/prepDE.py, accessed on 17 February 2025). All the statistical analyses were performed with R version 4.4.1 (https://www.r-project.org/, accessed on 18 February 2025) via trimmed mean of M values (TMM) normalization of the count data and the edgeR package, version 4.21. Using TCC-GUI version 1.0 [20], an MA plot was generated utilizing expressed genes, and DEGs with a false discovery rate (FDR) < 0.01 were extracted. Genes expressed in *P. memnon* were annotated for the reference transcriptome of *P. xuthus* version 1.1 obtained from the NCBI database using BLAST version 2.13.0+ (blastn).

### Real-time quantitative PCR

cDNA was synthesized from 500 ng of total RNA treated with DNase I (Invitrogen, Van Allen Way, Carlsbad, CA, USA), following RNA extraction from the larval midgut and fat bodies using a PrimeScript™ 1st strand cDNA Synthesis Kit (Takara Co. Ltd., Shiga, Japan) according to the manufacturer's instructions. Real-time quantitative PCR (RT–qPCR) was conducted in 20-µL reaction volumes containing 0.5 µL of cDNA template, 0.4 µM of gene-specific primers, and KAPA SYBR

Fast qRT–PCR Kit reagents (Nippon Genetics Co., Ltd., Tokyo, Japan). RT–qPCR was performed on a Step One Plus Real-Time PCR System (Applied Biosystems Foster City, CA) and analyzed via the ΔΔCt method. Relative quantification (RQ) values were calculated based on the expression of *rpL31* (ribosomal protein L31), employed as the endogenous control. Primer sequences are listed in <u>S1 Table</u>.

### Extraction from samples

The larval frass and *C. paradisi* leaves were freeze-dried via lyophilization (VD-250F, Taitec Co. Ltd., Saitama, Japan) for 24 hours. These freeze-dried samples in Erlenmeyer flasks were extracted with three volumes of n-hexane (Hex.; Fujifilm Wako Pure Chemical Corp., Osaka, Japan) for 24 hours. The extract was filtered to be transferred into a recovery flask, and the pellet was returned to the Erlenmeyer flask. Equal amounts of n-hexane were added to the Erlenmeyer flasks. The solvent in the recovery flask was evaporated using a rotary evaporator (N-1110 N; Tokyo Scientific Instruments Co., Ltd., Tokyo, Japan). These operations were repeated three times; thereafter, the samples were extracted with three volumes of chloroform (Chl.; Fujifilm Wako Pure Chemical Corp.) in the same way as the extraction with n-hexane. Finally, we performed the same procedure as that used for extraction with n-hexane but with methanol (Met.; Fujifilm Wako Pure Chemical Corp.). Each extract was collected in brown screw-top tubes (Maruemu Corp., Osaka, Japan), and the solvent remaining in the tubes was evaporated at room temperature in the dark. Hex. extract (262 mg), Chl. extract (387 mg), and Met. extract (2032 mg) were obtained from 53.5 g of freeze-dried frass. Hex. extract (20 mg), Chl. extract (13 mg), and Met. extract (769 mg) were obtained from 5.07 g of freeze-dried leaves.

### Cell culture

MIA PaCa2 (RCB2094) cells were obtained from the Riken Cell Bank (Riken, Tsukuba, Japan). These cells were cultured in Dulbecco's modified Eagle medium (DMEM; Nakalai Tesque, Inc., Kyoto, Japan) supplemented with 10% fetal bovine serum (FBS; Nichirei Biosciences, Inc., Tokyo, Japan) and one hundred units/mL penicillin–streptomycin (Fujifilm Wako Pure Chemical Corp.) in a T25 flask at 37°C in a humidified chamber containing 5% $CO_2$. To maintain the cells, 70–80% confluent cells were treated with TrypLE™ Express (Thermo Fisher Scientific) and passaged.

### Cell viability assay

MIA PaCa2 cells (1100 cells/well) were seeded in 96-well plates and cultured for 24 hours. The mixture of medium and samples dissolved in dimethyl sulfoxide (DMSO; Nakalai Tesque) was added to the cells in the plate. After 48 hours of incubation, 10 µL of WST-8 reagent (Dojin Chemical Co., Kumamoto, Japan) was added to the wells. After incubation for 4 hours, the absorbance at 450–470/620 nm was measured via a Gen5 microplate reader (Agilent Technologies) according to the manufacturer's instructions. The cell viability was calculated with the following formula after the absorbance at 620 nm was subtracted from the absorbance at 450–470 nm.

$$Cell\ viability\ (\%) = \frac{ABSsample - ABSblank}{ABScontrol - ABSblank} \times 100$$

### Cell morphological observation

MIA PaCa2 cells were seeded in 6-well plates (80000 cells/well) or 24-well plates (20000 cells/well) and cultured for 24 hours. After removing the medium in each well of 6-well plates, new medium with samples dissolved in DMSO

was added. Cisplatin (CDDP; Fujifilm Wako Pure Chemical Corp.) prepared according to the manufacturer's instructions was used as a positive control. The cells were cultured for 48 hours, and the cell morphology was observed using a microscope (CKX53; Olympus Co., Tokyo, Japan). Photos were taken with a FLEXCAM C1 (Leica, Wetzlar, Germany).

## Column chromatography

The Chl. extract from the larval frass (712.1 mg) dissolved in chloroform (3 mL) was applied to the top of a silica gel column (120 mm height) packed into a glass tube (30 × 300 mm). First, 100% chloroform was run through the column, and the solvent was collected in a recovery flask (Chl. Fr., 172.4 mg). Next, 100% ethyl acetate (E.A.; Fujifilm Wako Pure Chemical Corp.) was run through the column, and the solvent was collected in a recovery flask (E.A. Fr., 406.3 mg). Finally, 100% methanol was run through the column, and the eluent was collected in a recovery flask (Met. Fr., 70.5 mg). Each fraction collected in a recovery flask was transferred into brown screw-top tubes (Maruemu Corp.), and the solvent in each recovery flask was evaporated with a rotary evaporator. The screw tubes were stored at −30 °C until use. Met. Fr. (60.6 mg) dissolved in a mixed solvent (1 mL), chloroform:methanol (10:1, v/v), was applied to the top of a silica gel column (170 mm height) packed into a glass tube (30 × 300 mm). The mixed solvent was run through the column, and the eluent was collected in three glass test tubes (24 mL/tube) to obtain Fr. 1 (6.4 mg), Fr. 2 (2.5 mg), and Fr. 3 (5.9 mg), which exhibited some spots on thin-layer chromatography (TLC) plates. The fractions collected in glass test tubes were transferred to recovery flasks, and the solvent was removed using a rotary evaporator. Finally, the fractions were transferred to brown screw-top tubes (Maruemu Corp.). All experiments were conducted at room temperature (25°C).

## HPLC

To separate the compounds included in Fr. 1 by HPLC, Fr. 1 dissolved in methanol (20 mg/mL) was filtered with a 0.22 µm PVDF filter (Merck Millipore). HPLC was carried out via a Jasco system (Pu-4580 HPLC pump, UV-4070 UV–visible detector, MX-4580-0 dynamic mixing unit, DG-4580 degassing unit; Jasco, Gross-Umstadt, Germany) equipped with an XBridge® Prep OBD™ $C_{18}$ column (19 × 250 mm, particle size 10 µm; Waters Co., Milford, MA, USA). Then, 100% methanol (Fujifilm Wako Pure Chemical Corp.) was run through the column, and the eluates detected at 256 nm or 365 nm were obtained from 10 mg of Fr. 1 as Fr. 4 (5.2~8.4 min, 1.4 mg), Fr. 5 (8.4~11.5 min, 0.8 mg), Fr. 6 (11.5~14.0 min, 0.5 mg), Fr. 7 (14.0~17.6 min, 0.7 mg), Fr. 8 (17.6~19.6 min, 0.2 mg), Fr. 9 (19.6~21.1 min, 0.5 mg) and Fr. 10 (21.1~24.4 min, 0.5 mg) in glass test tubes. These fractions were transferred into recovery flasks, and the solvent was evaporated with a rotary evaporator. Finally, each fraction was transferred to brown screw-top tubes (Maruemu Corp.), and the solvent was removed using a rotary evaporator.

## Thin-layer chromatography (TLC)

Samples (the Chl. extract of the frass and the leaves, Chl. Fr., E.A. Fr., Met. Fr., and Fr. 1 to Fr. 10) dissolved in chloroform were mounted on a silica TLC plate (Silica gel 60 $F_{254}$; Merck Millipore, Burlington, MA, USA) using a glass capillary. The plate was developed with a mixed organic solvent, chloroform:methanol (10:1, v/v), at room temperature (25°C) until the solvent front reached 7 cm from the spot origin. The plate was dried, and the spots detected by UVA (366 nm) and UVC (256 nm) exposure were marked with a pencil.

## MALDI-Spiral TOF-TOF analysis

Fr. 7 and Phe-a (Cayman Chemical, Ann Arbor, MI, USA) dissolved in methanol were mixed with a MALDI matrix, α-cyano-4-hydroxycinnamic acid (CHCA; Fujifilm Wako Pure Chemical Corp.), saturated in methanol. Fr. 10 and Pph-a

(Cayman Chemical) dissolved in methanol were mixed with a MALDI matrix, *trans*-2-[3-(4- *tert*-butylphenyl)-2-methyl-2-propenylidene] malononitrile (DCTB; Tokyo Chemical Industry Co. Ltd., Tokyo, Japan), dissolved in acetonitrile (Fujifilm Wako Pure Chemical Corp.). These mixtures were applied to a MALDI plate, and the samples were analyzed with a JMS-S3000 Spiral TOF (Jeol Ltd., Tokyo, Japan) equipped with the TOF/TOF option. Mass tolerances for Fr. 7 ([M] $^+$ *m/z* 592.2673) and Phe-a standard ($C_{35}H_{36}N_4O_5$, [M] $^+$ *m/z* 592.2680), and for Fr. 10 ([M] $^+$ *m/z* 534.26245) and Pph-a ($C_{33}H_{34}N_4O_3$, [M] $^+$ *m/z* 534.26254) standard were −1.2 ppm and −0.17 ppm, respectively. The structural formulas of Phe-a and Pph-a were drawn using MarvinSketch version 22.2.0.

## LC-ESI-MS analysis

Fr. 7, Fr. 10, Phe-a, and Pph-a dissolved in methanol were injected into a Capcell Core $C_{18}$ column (2.1 × 150 mm, particle size 2.7 μm; Osaka Soda Co. Ltd., Osaka, Japan), detected using LC systems (SCL-10A VP system controller, LC-20AD HPLC pump, SPD-20A UV–visible detector, DGU-20A 5R degassing unit; Shimadzu Corp., Kyoto, Japan), and analyzed via MS (Orbitrap LTQ XL™; Thermo Fisher Scientific Inc.). A mixture of 0.1 vol% formic acid-acetonitrile:0.1 vol% formic acid-distilled water (95:5, v/v; Fujifilm Wako Pure Chemical Corp.) was run through the system at 0.2 mL/min. The peaks were detected at 366 nm. Mass tolerances for Fr. 7 ([M] $^+$ *m/z* 593.2756) and Phe-a standard ($C_{35}H_{36}N_4O_5$, [M] $^+$ *m/z* 593.2758), and for Fr. 10 ([M] $^+$ *m/z* 535.26999) and Pph-a standard ($C_{33}H_{34}N_4O_3$, [M] $^+$ *m/z* 535.2704) were −0.34 ppm and −0.97 ppm, respectively.

## NMR spectroscopy

The Fr. 7 yield reached 3.6 mg, and the Fr. 10 yield reached 3 mg, by repeated HPLC. The NMR spectra of the samples and standards (Pph-a and Phe-a) were measured using a 600 MHz NMR spectrometer (Jeol, Tokyo, Japan). Measurement conditions are listed in S2 Table. The chemical shifts were analyzed via Delta version 6.0.0 (Jeol). The NMR spectra and chemical structures are shown in S1–S3 Figs.

## Aβ 1–42 aggregation assay

Aβ1–42 aggregation can be investigated by measuring the fluorescence intensity of ThT, and EGCG inhibits the aggregation [21,22]. Amyloid beta-protein (Human, 1–42) (Peptide Institute, INC., Osaka, Japan) was dissolved in 0.5 mM hexafluoroisopropanol (HFIP; Fujifilm Wako Pure Chemical Corp.) at 0.5 mM and incubated on ice for 15 min with gentle agitation. After dispensing in Protein LoBind tubes (Eppendorf, Hamburg, Germany), we removed HFIP with a centrifugal concentrator at room temperature for 15 min to allow the Aβ1–42 film to form. The Aβ1–42 film was stored at −30°C until use. Thioflavin T (ThT; Sigma–Aldrich, St. Louis, MO, USA) was dissolved in 0.1 N HCl at a concentration of 0.5% (w/v) and diluted with PBS to 80 μM. Phe-a and Pph-a were dissolved in DMSO at 1 mM, and epigallocatechin gallate (EGCG; Tokyo Chemical Industry Co., Ltd.) as a positive control was dissolved in DMSO at 10 mM. Each sample was diluted with PBS at 400 μM (EGCG) or 40 μM (Phe-a and Pph-a). The Aβ1–42 film was dissolved in DMSO at 1 mM, sonicated for 30 s and incubated for 30 s at room temperature; this process was repeated five times via a Bioruptor® II (Sonic Bio Co. Ltd., Kanagawa, Japan). The Aβ1–42 solution was diluted with PBS at 40 μM. A mixture of 20 μM Aβ1–42, 20 μM ThT, and 100 μM EGCG or 10 μM Phe-a or Pph-a was prepared. We prepared a mixture without Aβ1–42 as a control and a mixture without Aβ1–42 and ThT as a blank. One hundred microliters of the mixture was added to a 96-well black plate, and fluorescence was measured using a Gen5 instrument (excitation; 450 nm, emission; 482 nm) at 0 min, 30 min, 60 min and 90 min with incubated at 37°C. The fluorescence intensity of the mixture with or without Aβ1–42 was calculated by subtracting the value obtained for the blank.

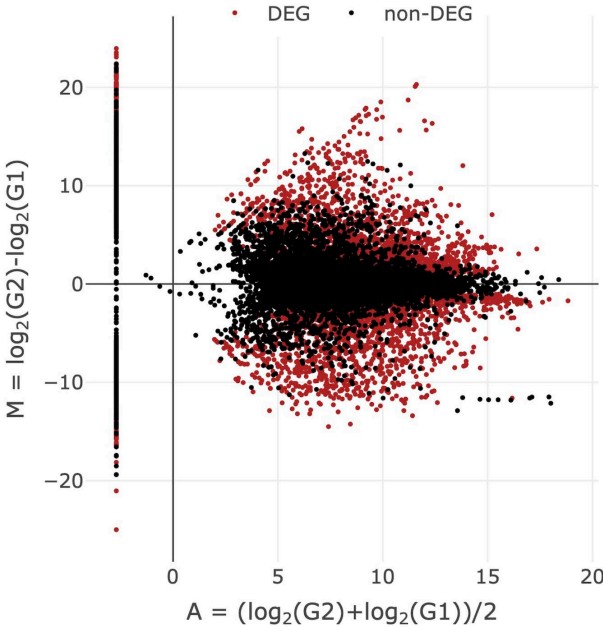

**Fig 1. Comparison of gene expression between *C. paradisi*-fed larval fat bodies and the midgut.** A total of 5290 of the 17858 total DEGs had a false discovery rate (FDR)<0.01.

## Results

### Comparison of expressed genes between the midgut and fat bodies

The expression of genes encoding metabolic enzymes was analyzed by RNA sequencing analysis. Genes expressed in the midgut and fat bodies were compared by edgeR to examine the organs related to the metabolism of components from *C. paradisi* in *P. memnon* larvae. A total of 17858 genes were expressed in the midgut and fat bodies of *P. memnon* larvae, 5290 of which were differentially expressed genes (DEGs) (Fig 1 and S3 Table). Annotation of these DEGs against *P. xuthus* reference transcripts led to the identification of 35 genes encoding metabolic enzymes, *cytochrome P450*s (*CYP*s), *glutathione S-transferase*s (*GST*s), a *carboxylesterase* (*CCE*), and an *epoxide hydrolase* (*EH*) (Tables 1 and 2 and S4 Table). Seventeen *CYP*s, *GST1*, and *EH3* were upregulated in the midgut, whereas 13 *CYP*s, *GST1* (isoform D), and *CCE3* were upregulated in the fat bodies (Tables 1 and 2). In particular, 15 genes in the midgut and fat bodies were annotated as CYP6B enzymes, which are involved in metabolizing host plant components in Papilionidae larvae [23,24]. Among these genes, 10 were upregulated in the midgut, whereas 5 were upregulated in fat bodies (Tables 1 and 2).

Because *CYP*s involved in metabolizing host plant components are more highly expressed in the larval midgut than in the larval fat bodies [25], the expression levels of *CYP6B2* (MSTRG.2586.1), *CYP6B4* (MSTRG.9993.1), and *CYP6B5* (MSTRG.8684.1) were validated in both tissues via RT–qPCR. These *CYP*s showed higher expression in the midgut than in the fat bodies, consistent with the RNA sequencing results (S4 Fig).

These results indicate that *C. paradisi* leaf components are primarily metabolized in the midgut of *P. memnon* larvae and excreted into larval frass.

                                                                                

**Table 1. TPMs of metabolic enzyme genes upregulated in the midgut compared with the fat bodies.**

| P. memnon | P. xuthus | | TPM | |
|---|---|---|---|---|
| Transcript IDs | Transcript IDs | Transcript Name | MG | FB |
| MSTRG.1986.1 | XM_013306334.1 | CYP6B2-like (LOC106113517) | 4.17 | 0.0198 |
| MSTRG.1987.1 | XM_013306334.1 | CYP6B2-like (LOC106113517) | 6.51 | 0 |
| MSTRG.1989.1 | XM_013306337.1 | CYP6B2-like (LOC106113520) | 27.7 | 0.194 |
| MSTRG.2586.1 | XM_013316774.1 | CYP6B2-like (LOC106121230) | 356 | 20.1 |
| MSTRG.9994.1 | XM_013314216.1 | CYP6B4-like (LOC106119301) | 20.6 | 0.0516 |
| MSTRG.9993.1 | XM_013314216.1 | CYP6B4-like (LOC106119301) | 207 | 1.36 |
| MSTRG.9995.1 | XM_013314217.1 | CYP6B5-like (LOC106119302) | 343 | 0.663 |
| MSTRG.8684.1 | XM_013323145.1 | CYP6B5-like (LOC106125793) | 875 | 11.3 |
| MSTRG.1984.2 | XM_013306304.1 | CYP6B6-like (LOC106113499) | 8.83 | 0.374 |
| MSTRG.1985.1 | XM_013306304.1 | CYP6B6-like (LOC106113499) | 590 | 119 |
| MSTRG.11274.3 | XM_013318339.1 | CYP6k1-like (LOC106122393) | 3.89 | 0.00870 |
| MSTRG.11274.2 | XM_013318339.1 | CYP6k1-like (LOC106122393) | 5.72 | 0.0138 |
| MSTRG.11469.1 | XM_013318339.1 | CYP6k1-like (LOC106122393) | 195 | 0.908 |
| MSTRG.11273.1 | XM_013318339.1 | CYP6k1-like (LOC106122393) | 224 | 1.88 |
| MSTRG.955.1 | XM_013324930.1 | CYP6j1-like (LOC106127006) | 13.7 | 1.74 |
| MSTRG.11275.1 | XM_013318379.1 | Probable CYP6w1 (LOC106122430) | 14.9 | 0.0225 |
| MSTRG.12902.1 | XM_013306187.1 | CYP12A2-like (LOC106113436) | 37.7 | 7.71 |
| MSTRG.206.1 | XM_013310872.1 | GST1-like (LOC106116829) | 2.97 | 0 |
| MSTRG.7461.1 | XM_013319544.1 | Epoxide hydrolase 3-like (LOC106123298) | 195 | 8.44 |

MG; midgut, FB; fat body, *CYP*; *Cytochrome P450, GST*; *Glutathione S-transferase*.

**Table 2. TPMs of metabolic enzyme genes upregulated in the fat bodies compared with the midgut.**

| P. memnon | P. xuthus | | TPM | |
|---|---|---|---|---|
| Transcript_IDs | Transcript_IDs | Transcript_Name | MG | FB |
| MSTRG.12333.1 | XM_013321418.1 | CYP4C1-like (LOC106124584), transcript variant X2 | 0 | 22.0 |
| MSTRG.11684.1 | XM_013321418.1 | CYP4C1-like (LOC106124584), transcript variant X2, mRNA | 0.0234 | 28.4 |
| MSTRG.3668.1 | XM_013321417.1 | CYP4C1-like (LOC106124584), transcript variant X1 | 0.340 | 42.1 |
| MSTRG.13644.1 | XM_013316294.1 | CYP6B1 (LOC106120835) | 0 | 11.3 |
| MSTRG.9999.1 | XM_013314221.1 | CYP6B4-like (LOC106119305) | 6.03 | 21.5 |
| MSTRG.10913.1 | XM_013318668.1 | CYP6B5-like (LOC106122600) | 0 | 6.83 |
| MSTRG.9996.1 | XM_013314222.1 | CYP6B5-like (LOC106119306) | 0.292 | 23.9 |
| MSTRG.9998.1 | XM_013314218.1 | CYP6B5-like (LOC106119303) | 3.33 | 353 |
| MSTRG.11276.1 | XM_013318318.1 | CYP6k1-like (LOC106122377) | 0.104 | 10.4 |
| MSTRG.8383.1 | XM_013311539.1 | CYP6k1-like (LOC106117299) | 4.86 | 50.1 |
| MSTRG.6061.1 | XM_013316465.1 | CYP9e2-like (LOC106120994) | 0.0432 | 23.9 |
| MSTRG.3229.1 | XM_013309630.1 | Probable CYP304a1 (LOC106115964) | 121 | 285 |
| MSTRG.3228.1 | XM_013309630.1 | Probable CYP304a1 (LOC106115964) | 228 | 528 |
| MSTRG.10357.1 | XM_013307499.1 | GST1, isoform D-like (LOC106114338) | 9.67 | 24.9 |
| MSTRG.9257.1 | XM_013306094.1 | CCE3 (LOC106113370) | 1.86 | 7.59 |

MG; midgut, FB; fat body, *CYP*; *Cytochrome P450, GST*; *Glutathione S-transferase, CCE*; *Carboxylesterase*.

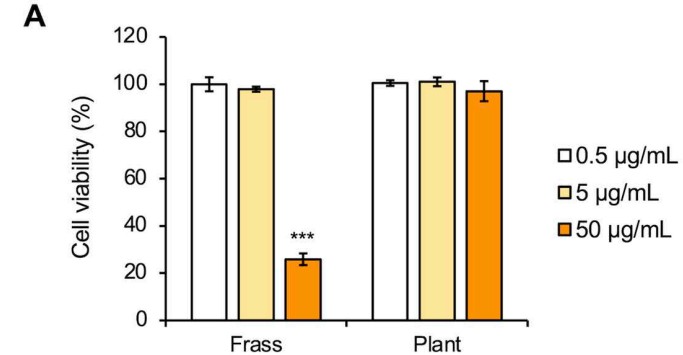

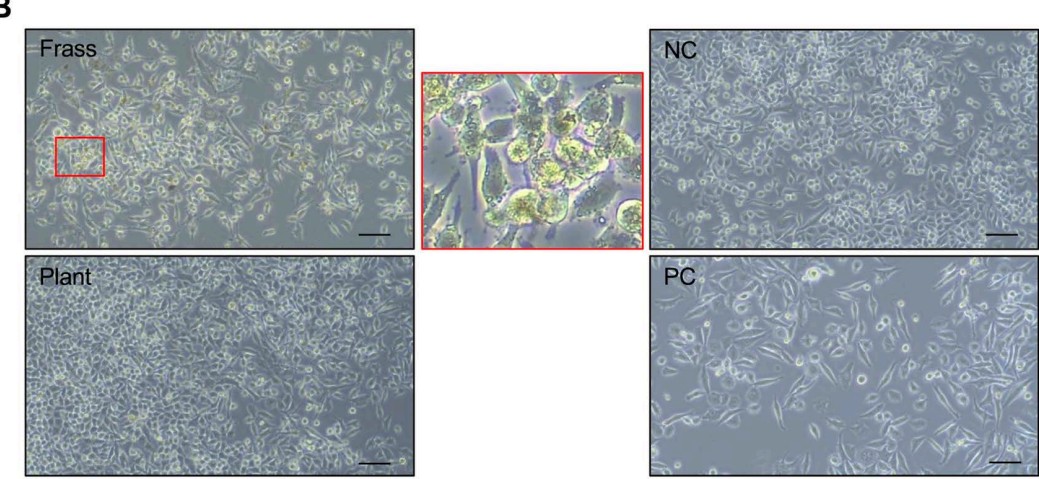

**Fig 2. Effects of the chloroform extract on MIA PaCa2 cells. (A)** Cell viability relative to that of the control treatment with 0.1% (v/v) dimethyl sulfoxide (DMSO) after treatment with the Chl. extract from the larval frass and *C. paradisi* leaves (plant), as determined via the WST-8 assay. The error bars represent the means±SDs from three biological replicates. Significant differences from the control were analyzed with Student's t test, ***P<0.001. **(B)** Cell morphological observations after treatment with the Chl. extract from the larval frass and *C. paradisi* leaves at 50 µg/mL, with 0.1% (v/v) DMSO (negative control; NC) and 10 µM CDDP (positive control; PC) as controls. The image in the red square is an enlarged view. Scale bars=100 µm.

### Effect of extracts from larval frass and *C. paradisi* on MIA PaCa2

The components from *C. paradisi* leaves metabolized by the midgut of *P. memnon* larvae may be included in the larval frass. Thus, we examined the biological activity of components extracted from *C. paradisi* leaves and *C. paradisi*-fed larval frass, and we compared the effects of the extracts on the human pancreatic cancer cell line MIA PaCa2. First, the components from larval frass and *C. paradisi* leaves were extracted using hexane (Hex.), chloroform (Chl.), or methanol (Met.), depending on their polarity. Next, the viability of MIA PaCa2 cells after treatment with these extracts from larval frass and *C. paradisi* was investigated. While the Chl. extract from larval frass significantly inhibited the viability of MIA PaCa2 cells (25.9%) at 50 µg/mL, the Chl. extract from *C. paradisi* leaves did not reduce the cell viability (97.0%) (Fig 2A). Furthermore, the Chl. extract from the larval frass induced changes in cell morphology compared with the Chl. extract from *C. paradisi* leaves and the negative control (Fig 2B). However, the Hex. and Met. extracts from both the larval frass and *C. paradisi* leaves did not reduce cell viability (S5A and S5B Fig). Furthermore, these extracts did not induce cell morphological changes (S5C and S5D Fig). Therefore, the

components and their biological activities differed between the frass from *C. paradisi*-fed *P. memnon* larvae and *C. paradisi* leaves.

## Separation of the active components from larval frass

The components included in the Chl. extract from larval frass and *C. paradisi* leaves were compared via TLC. Seven spots with Rf values less than 0.58 were detected only in the extract from the larval frass compared with the extract from *C. paradisi* leaves (Fig 3A, arrowheads).

To separate the bioactive components from Chl. extract of the larval frass, we conducted open-column chromatography with Chl., E.A., and Met. First, the Chl. extract from the larval frass was separated by elution with Chl., E.A., and Met. in that order; the eluates were collected as the Chl. fraction (Fr.), E.A. Fr., and Met. Fr. These fractions contained compounds with differing polarities; Chl. Fr. had a spot with an Rf value of 0.96, E.A. Fr. had spots with Rf values ranging from 0.37 to 0.97, and Met. Fr. had spots with Rf values ranging from 0.086 to 0.57 according to TLC analysis (Fig 3B). Met. Fr. at 50 µg/mL significantly reduced the viability of MIA PaCa2 cells to 32.7%, and the Chl. Fr. and E.A. Fr. did not decrease cell viability (S6A Fig). Furthermore, Met. Fr. significantly decreased the cell density compared with Chl. Fr., E.A. Fr., and the negative control (S6B Fig).

Next, Met. Fr. was separated using a mixture of the solvents Chl. and Met. to obtain three fractions (Fr. 1 to Fr. 3). These three fractions at 50 µg/mL significantly inhibited the viability of MIA PaCa2 cells to 7.3%, 4.9%, and 17.3%, respectively. In particular, compared with Fr. 2 (46.6%) and Fr. 3 (103%), Fr. 1 strongly reduced the cell viability at 5 µg/mL to 22.6% (S6C Fig). Moreover, the cell density of MIA PaCa2 cells was decreased by treatment with 50 µg/mL Fr. 1 or Fr. 2 (S6D Fig). Among the components of these fractions, Fr. 1 had an apparent black spot with an Rf value of 0.43 compared with those of Fr. 2 and Fr. 3 (Fig 3C, arrowhead).

Finally, Fr. 1 was separated by HPLC to yield seven fractions, Fr. 4 to Fr. 10 (Fig 3D and 3E and S7 Fig). Fr. 4, Fr. 7, and Fr. 10 at 40 µg/mL reduced the viability of MIA PaCa2 cells by 49.0%, 61.2%, and 66.8%, respectively (Fig 4A). Furthermore, compared with the negative control, Fr. 4 significantly decreased the cell density, and Fr. 10 induced

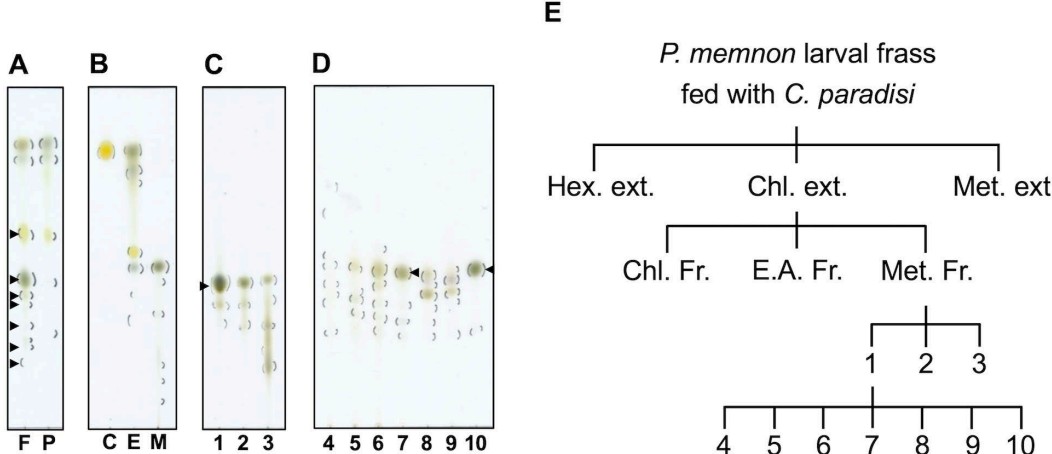

**Fig 3. Analysis of components and fractionation. (A-D)** Comparison of components by TLC. **(A)** The Chl. extracts of the larval frass (lane F) and *C. paradisi* leaves (lane P). Arrow heads show the only spots detected in lane F. **(B)** Chl. Fr. (lane C), E.A. Fr. (lane E), and Met. Fr. (lane M) separated from the Chl. extract of the larval frass by column chromatography. **(C)** Fr. 1 to Fr. 3 separated from Chl. Fr. by column chromatography. The arrowhead shows the specific spot detected in Fr. 1. **(D)** Fr. 4 to Fr. 10 separated from Fr. 1 by HPLC. The arrowheads show the colored compounds contained in Fr. 7 and Fr. 10. **(E)** Fractionation of the Chl. extract from the larval frass by column chromatography and HPLC.

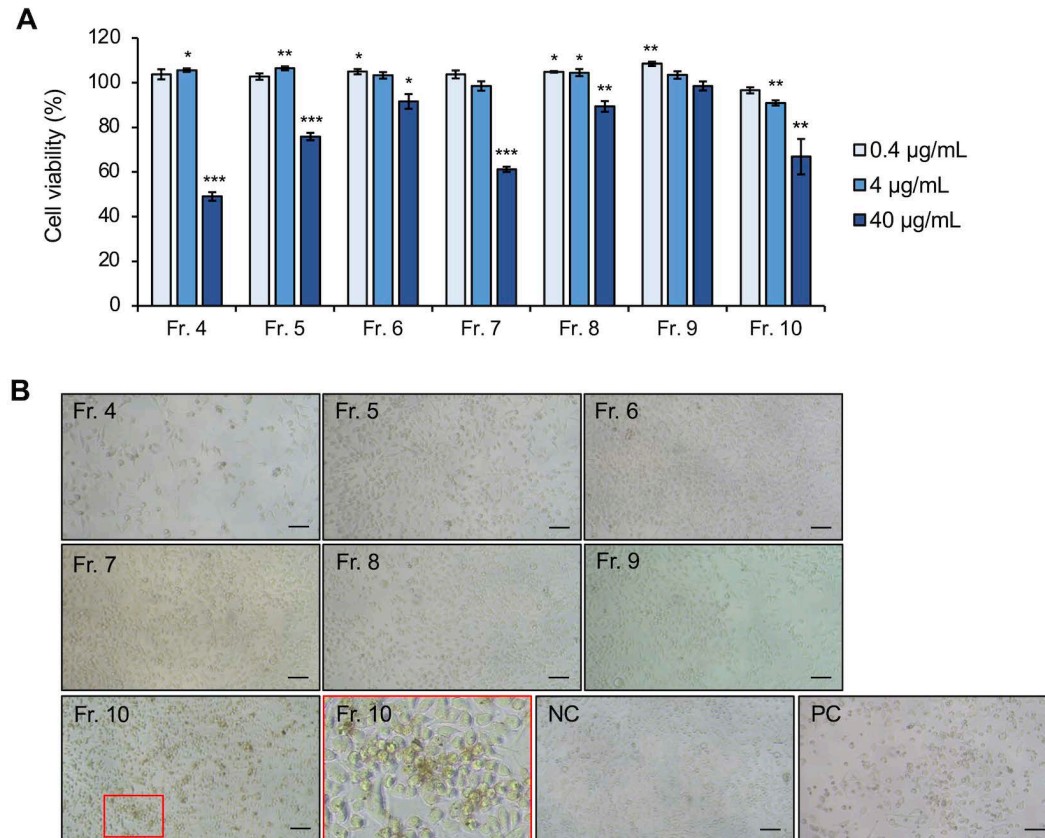

**Fig 4. Effects of Fr. 4 to Fr. 10 on the viability and morphology of MIA PaCa2 cells. (A)** Cell viability relative to that of the control treatment with 0.1% (v/v) DMSO after treatment with Fr. 4 to Fr. 10, as determined by a WST-8 assay. The error bars represent the means ± SDs from three biological replicates. Significant differences from the control were analyzed with Student's t test; *P < 0.05, **P < 0.01, ***P < 0.001. **(B)** Cell morphology observation after treatment with Fr. 4 to Fr. 10 at 40 μg/mL, with 0.1% (v/v) DMSO (NC) and 10 μM CDDP (PC) as controls. The image in the red square is an enlarged view. Scale bars = 100 μm.

morphological changes similar to those in cells after treatment with Chl. extract from the larval frass (Figs 2B and 4B). While Fr. 4 contained some UV-responsive compounds, Fr. 7 and Fr. 10 included colored compounds with an Rf value of 0.54 (Fig 3D, arrowheads). Therefore, the Chl. extract from the larval frass might contain some compounds that inhibit cell viability and induce morphological changes in MIA PaCa2 cells.

### Identification of the compounds included in Fr. 7 and Fr. 10

Fr. 4, Fr. 7, and Fr. 10 inhibited the viability of MIA PaCa2 cells, and fewer spots were observed with Fr. 7 and Fr. 10 than with Fr. 4 by TLC (Fig 3D). Therefore, the compounds included in Fr. 7 and Fr. 10 were analyzed via MS. Since the predicted molecular weight of Fr. 7 and Fr. 10 was 592 (S8A Fig) and 534 (S8B Fig), it was estimated that the color of these fractions was derived from the chlorophyll-a (Chl-a) catabolites pheophoebide-a (Phe-a, $C_{35}H_{36}N_4O_5$) and pyropheophorbide-a (Pph-a, $C_{33}H_{34}N_4O_3$). Thus, to identify the main components of Fr. 7 and Fr. 10, these fractions were compared with the standards of Phe-a and Pph-a by MS/MS using MALDI-Spiral TOF-TOF, LC-ESI-MS, and NMR.

Each product ion spectrum of Fr. 7 matched well with that of the Phe-a standard in MS/MS analysis (Fig 5A). The mass spectrum corresponded to Fr. 7 and Phe-a standard at 2.9 min in LC-ESI-MS analysis (S9A Fig). In addition, the [1]H

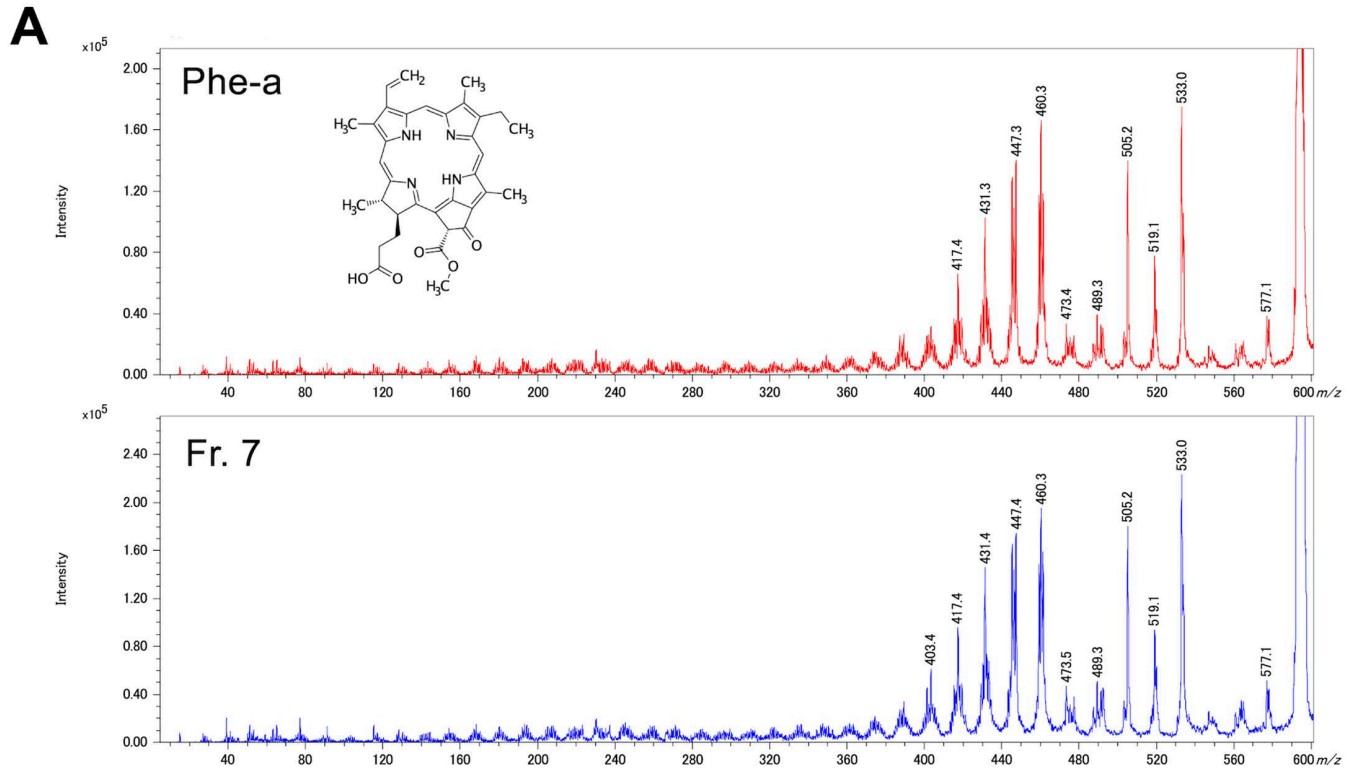

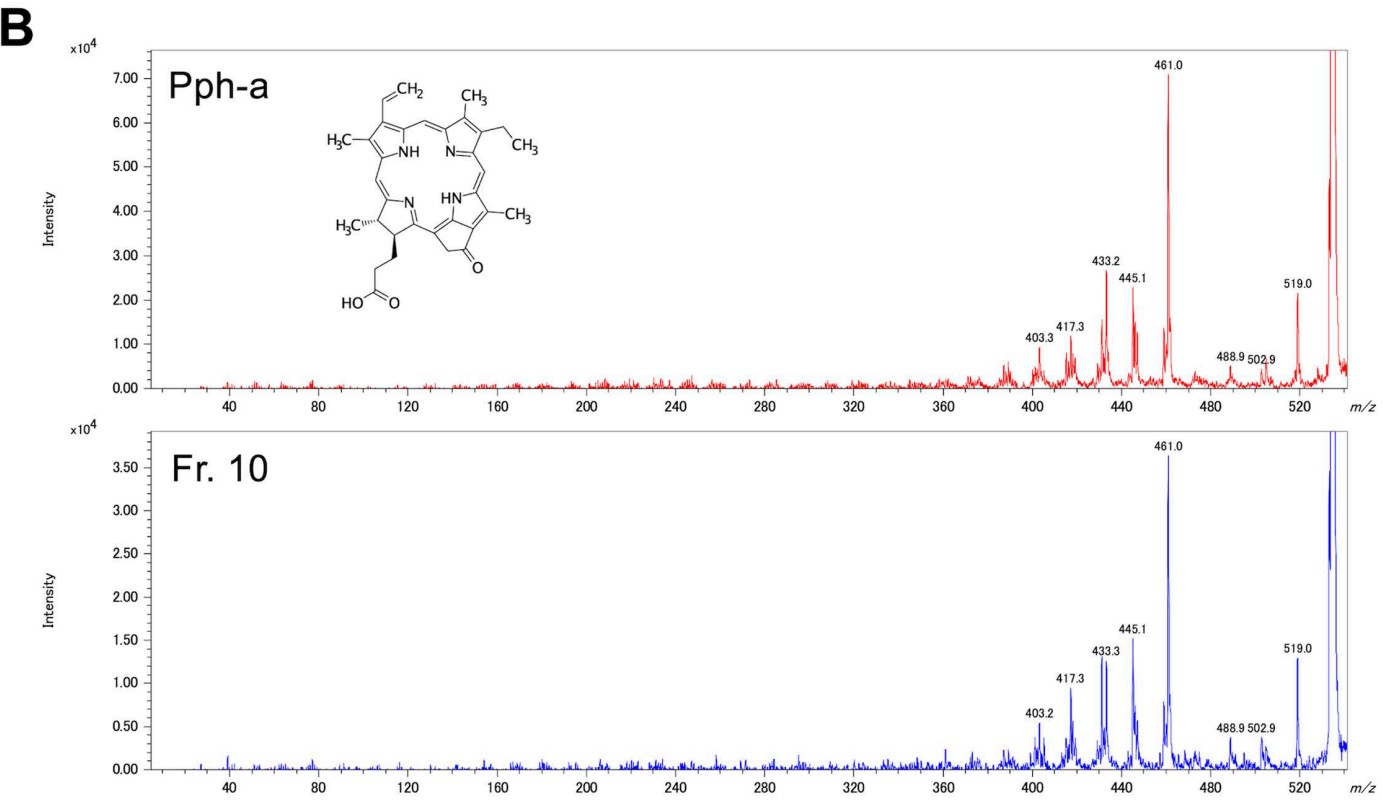

**Fig 5. Comparison of MS/MS data between the fractions and the standards by MALDI-Spiral TOF/TOF in positive-ion mode. (A)** The product ion spectra of Fr. 7 and Phe-a, with the selected precursor ion [M] $^{+\cdot}$ *m/z* 593.5. **(B)** The product ion spectra of Fr. 10 and Pph-a, with the selected precursor ion [M] $^{+\cdot}$ *m/z* 534.3.

**Table 3. Comparison of ¹H-NMR data between Fr. 10 and the Pph-a standard.**

| Fr. 10 | Pph-a standard |
|---|---|
| 9.38 (1H, s) | 9.42 (1H, s) |
| 9.29 (1H, s) | 9.31 (1H, s) |
| 8.50 (1H, s) | 8.52 (1H, s) |
| 7.93 (1H, m) | 7.95 (1H, dd, 17.9 Hz) |
| 6.24 (1H, d, 17.2 Hz) | 6.25 (1H, d, 17.9 Hz) |
| 6.13 (1H, d, 11.0 Hz) | 6.14 (1H, d, 11.0 Hz) |
| 5.23 (1H, d, 18.6 Hz) | 5.24 (1H, d, 19.2 Hz) |
| 5.08 (1H, d, 18.6 Hz) | 5.10 (1H, d, 19.2 Hz) |
| 4.44 (1H, s) | 4.46 (1H, t, 7.2 Hz) |
| 4.26 (1H, s) | 4.29 (1H, d, 9.6 Hz) |
| 3.60 (2H, m) | 3.63 (2H, t, 7.9 Hz) |
| 3.36 (3H, s) | 3.37 (3H, s) |
| 3.18 (3H, s) | 3.24 (3H, m) |
| 2.65 (2H, m) | 2.65 (2H, m) |
| 2.35 (1H, s) | 2.35 (1H, d, 9.6 Hz) |
| 2.23 (1H, s) | 2.25 (1H, s) |
| 1.78 (3H, d, 6.2 Hz) | 1.80 (3H, d, 6.9 Hz) |
| 1.65 (3H, s) | 1.66 (3H, t, 7.9 Hz) |

s, singlet; d, doublet; t, triplet; m, multiplet.

chemical shifts of Fr. 7 in the NMR spectrum matched those observed for the Phe-a standard and the literature data (Liu et al., 2022) for Phe-a (S5 Table).

Similarly, each product ion spectrum of Fr. 10 coincided with that of the Pph-a standard in MS/MS analysis (Fig 5B). The mass spectrum matched between Fr. 10 and the Pph-a standard at 7.1 min in LC-ESI-MS analysis (S9B Fig). Additionally, the ¹H and ¹³C chemical shifts of Fr. 10 matched well with those of the Pph-a standard (Tables 3 and 4).

Therefore, the main components of Fr. 7 and Fr. 10 were Phe-a and Pph-a, respectively.

## Examination of biological activity of Phe-a and Pph-a

The chemicals included in Fr. 7 and Fr. 10 were presumed to be Phe-a and Pph-a, respectively; we examined whether the biological activity of these fractions on MIA PaCa2 cells corresponded to that of the Phe-a and Pph-a standards. Phe-a did not affect cell viability and did not cause changes in cell morphology; however, Pph-a reduced the cell viability at 4 µg/mL and 40 µg/mL (Fig 6A) and induced changes in cell morphology, as well as Fr. 10 (Figs 4B and 6B). Considering these results, the main compound included in Fr. 7 was Phe-a; however, the bioactive substance included in Fr. 7 might be Phe-a. On the other hand, the main compound and the bioactive substance of Fr. 10 were predicted to be Pph-a.

To investigate the other biological activities of Phe-a and Pph-a, the effects of these compounds on amyloid β-protein (human, 1–42; Aβ1–42) aggregation were evaluated. Since ThT emits fluorescence in the presence of amyloid fibrils, the

**Table 4. Comparison of the $^{13}$C-NMR data between Fr. 10 and the Pph-a standard.**

| Fr. 10 | Pph-a standard |
|---|---|
| 196.5 | 196.5 |
| 177.0 | 177.3 |
| 171.4 | 171.4 |
| 160.2 | 160.2 |
| 155.2 | 155.3 |
| 150.7 | 150.8 |
| 149.0 | 149.0 |
| 145.0 | 145.0 |
| 141.6 | 141.6 |
| 137.8 | 137.8 |
| 136.2 | 136.2 |
| 136.0 | 136.0 |
| 135.8 | 135.8 |
| 131.6 | 131.6 |
| 130.3 | 130.3 |
| 129.2 | 129.2 |
| 128.3 | 128.3 |
| 122.5 | 122.5 |
| 105.9 | 106.0 |
| 104.0 | 104.1 |
| 97.1 | 97.1 |
| 93.0 | 93.0 |
| 51.5 | 51.5 |
| 50.0 | 50.0 |
| 48.0 | 48.0 |
| 30.7 | 30.6 |
| 29.7 | 29.6 |
| 23.1 | 23.1 |
| 19.4 | 19.4 |
| 17.4 | 17.4 |
| 12.1 | 12.1 |
| 12.0 | 12.0 |
| 11.2 | 11.2 |

fluorescence intensity of the mixture of ThT, Aβ 1–42, and Phe-a or Pph-a was assessed. While Aβ1–42 alone gradually increased the fluorescence intensity to $22.6 \times 10^3$ after 1.5 hours, 100 μM EGCG gradually decreased the fluorescence intensity to $1.9 \times 10^3$ after 1.5 hours. Despite the lower concentrations of Phe-a and Pph-a than those of EGCG, these standards at 10 μM gradually reduced the fluorescence, with the fluorescence intensity reaching $9.2 \times 10^3$ and $7.6 \times 10^3$, respectively, after 1.5 hours (Fig 6C).

## Discussion

Plants use secondary metabolites to protect themselves from predators or pathogens [26]. Herbivorous insects take up plant secondary metabolites with nutrients and metabolize these xenobiotics using their metabolic enzymes, namely, CYPs, CCEs, GSTs, and UDP-glycosyltransferases (UGTs) [27]. Insects are involved in a coevolutionary arms race

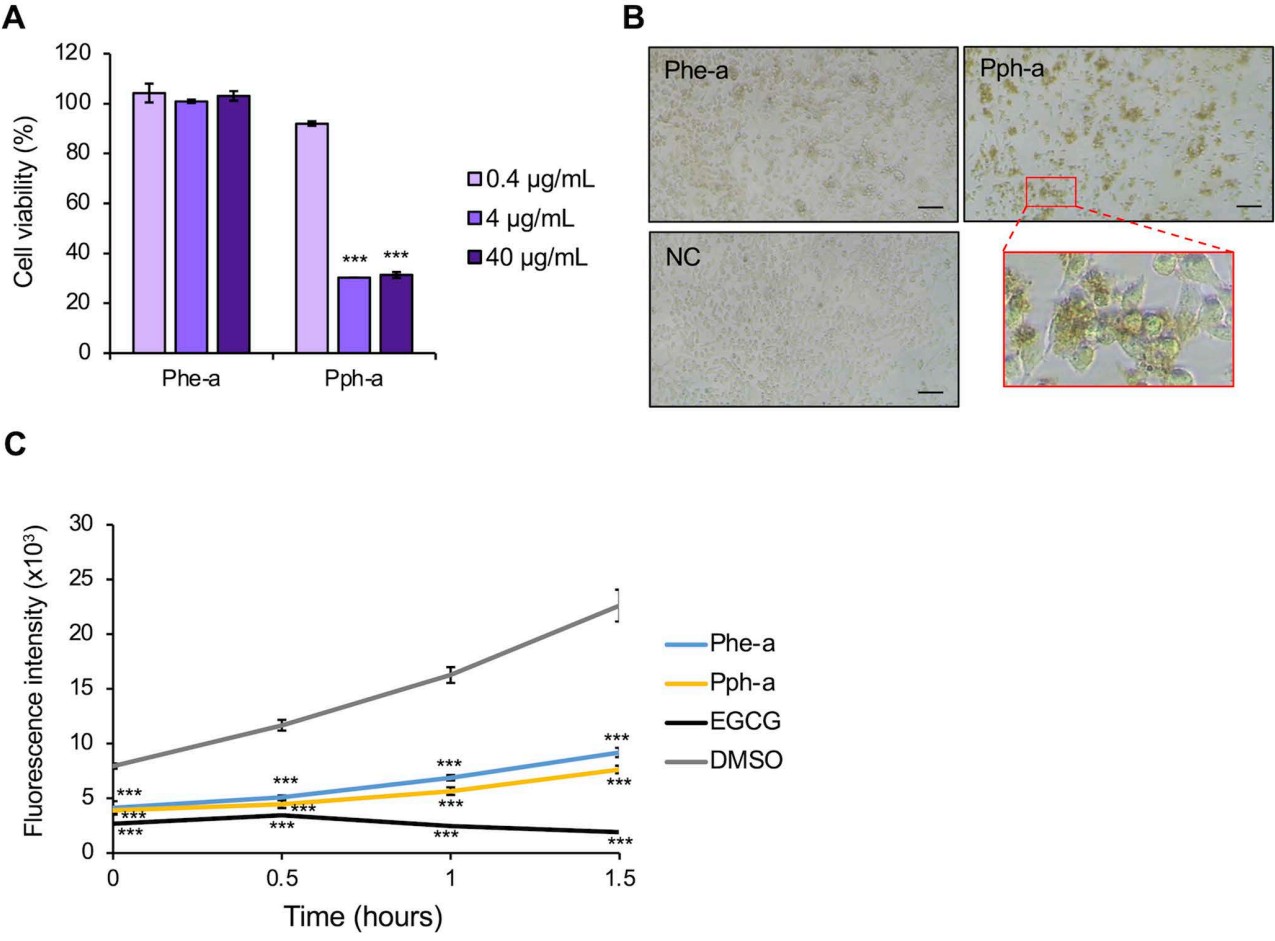

**Fig 6. Biological activity of Phe-a and Pph-a. (A)** Cell viability relative to the control treatment with 0.1% (v/v) DMSO after treatment with Phe-a and Pph-a. **(B)** Cell morphological observation after treatment with Phe-a and Pph-a at 40 µg/mL in 0.1% (v/v) DMSO (NC). The image in the red square is an enlarged view. Scale bars = 100 µm. **(C)** Effects of Phe-a, Pph-a, DMSO (as a negative control), and EGCG (as a positive control) on Aβ1-42 aggregation using ThT. Significant differences from the control were analyzed with Student's t test, ***P < 0.001.

with plants, and the ability of insects to metabolize plant chemicals affects host plant selection [28]. The *CYP6B* family plays a crucial role in the metabolism of plant secondary metabolites, namely, furanocoumarins with phototoxicity, and the inhibition of DNA replication in Papilionidae larvae [29]. *CYP6B1* and *CYP6B3* are upregulated in the midgut and fat bodies of *Papilio polyxenes* larvae, whose hosts are Apiaceae plants contain furanocoumarins derived via excessive intake of xanthotoxin [25]. Expression of the *CYP6B4* and *CYP6B17* genes in the larvae of the polyphagous Papilionidae insects *Papilio glaucus* and *Papilio canadensis* is induced by excessive consumption of xanthotoxin [30]. In addition to these CYP enzymes, *CYP6B21* from *P. glaucus* and *CYP6B25* from *P. canadensis* metabolize furanocoumarins [23]. Considering the chemical structure of xanthotoxin metabolites included in the larval frass of *P. polyxenes* and *Papilio multicaudatus* and the catalytic reaction of CYPs, the furan ring of furanocoumarins is epoxidated for ring opening [31,32]. Therefore, furanocoumarins may be metabolized in the *P. memnon* larval midgut because *CYP6B* and *EH3* transcripts were upregulated in the midgut of larvae that fed on *C. paradisi* leaves containing furanocoumarins (Tables 1 and 2).

Chlorophyll, a primary metabolite for photosynthesis, is broken down into chlorophyllide by chlorophyllase and ultimately into Phe-a and Pph-a during plant senescence [33]. However, Phe-a and Pph-a may also be produced from chlorophyll via the metabolic systems of *Spodoptera littoralis*, *Spodoptera eridania*, *Helicoverpa virescens*, *Helicoverpa armigera*, *Manduca sexta*, and *B. mori* [34,35], with chlorophyll metabolism likely occurring in the larval midgut [36]. Additionally, chlorophyllide binds to three proteins, i.e., red fluorescent protein (RFP), chlorophyllide binding protein (CHBP), and P252 protein, in the midgut of *B. mori* larvae [37–39]. Notably, an enzyme functioning as chlorophyllase has not been identified in insects, and it remains unclear whether RFP, CHBP, and P252 metabolize chlorophyllide into Phe-a and Pph-a. Thus, we speculated that *P. memnon* larvae metabolize Chl-a to Phe-a and Pph-a in the midgut via several metabolic enzymes that we identified by transcriptomic analysis in this study. We found that *CYP6B2*, *CYP6B4*, and *CYP6B5* mRNA expression was higher in the larval midgut than in the fat bodies. Therefore, these CYPs may play a role in the metabolism of Chl-a into Phe-a and Pph-a.

We speculate that Pph-a content may be higher in the frass than in the leaves, as the chloroform extract of the frass significantly reduced MIA PaCa2 cell viability (Fig 2A). Additionally, *S. littoralis* larval frass contained higher levels of Phe-a and Pph-a compared with the chlorophyll content in host plants [34]. These findings suggest that more Phe-a and Pph-a are produced from chlorophyll in host plants through metabolic processes in herbivorous insects.

As new biological discoveries, we found that Pph-a, which is present in the frass of *P. memnon* larvae fed *C. paradisi* leaves, inhibited the viability of MIA PaCa2, and Phe-a and Pph-a inhibited the aggregation of Aβ1–42, which is one of the causative factors in Alzheimer's disease [21,22].

The biological activities of Phe-a and Pph-a have been studied; a human promyelocytic leukemia cell line, HL-60, is induced to undergo apoptosis upon treatment with Phe-a or Pph-a and light, and Phe-a and Pph-a exhibit antiadipogenic activity [35,40]. Moreover, Phe-a, Pph-a, and their analogs are promising photosensitizers (PSs) for photodynamic therapy (PDT), providing approaches for antibacterial and anticancer treatment. For PDT, malignant tissues and bacteria with PSs are irradiated at a specific wavelength corresponding to the absorption maximum of the PS, leading to the generation of reactive oxygen species and further to cell death [41]. The conjugation of Phe-a with cancer-targeting moieties induces apoptosis in specific cell lines [42]. A Pph-a derivative has a strong cytotoxic effect on the human breast cancer cell line MDA-MB-231, which overexpresses *carbonic anhydrase IX* both in vitro and in vivo [43]. Furthermore, a conjugate of Pph-a with a cyclic cRGDfK peptide, with high binding affinity for integrin receptors overexpressed in many types of tumor cells or tumor vessels compared with normal cells, a highly hydrophilic polyethylene glycol chain and an extra strongly hydrophilic carboxylic acid inhibits the viability of some human cancer cell lines in vitro and reduces tumor volume in mice in vivo [44]. This study newly revealed that Phe-a and Pph-a inhibit Aβ1–42 aggregation. Elucidating their physiological mechanism using various mouse models is expected to clarify the effects of Phe-a and Pph-a.

CYPs are utilized for the synthesis of compounds [45]. Compared with mammalian CYPs, CYP109A2 and CYP109E1 from *Bacillus megaterium* more efficiently catalyze vitamin $D_3$ hydroxylation to generate the biologically active form, which is important for the regulation of calcium and phosphate metabolism. These CYPs have potential industrial applications [46,47]; however, the reaction of bacterial and mammalian *CYP*s requires heating to 37°C. In contrast, the CYP6B1 and CYP6B3 enzymes from *P. polyxenes*, expressed using insect cell systems and a baculovirus expression vector, require heating to only 30°C [24]. Therefore, the metabolic capacity of *P. memnon* larvae might be useful for bioprocessing Phe-a and Pph-a from chlorophyll, and our findings could support the development of new bioprocess techniques with low environmental impact using insect CYPs.

In conclusion, we found that *P. memnon* larvae take up Chl-a from *C. paradisi* leaves, after which the chemical structure of Chl-a changes to Phe-a and Pph-a through the metabolic ability of *P. memnon*, after which it is excreted with the frass of *P. memnon* larvae (Fig 7).

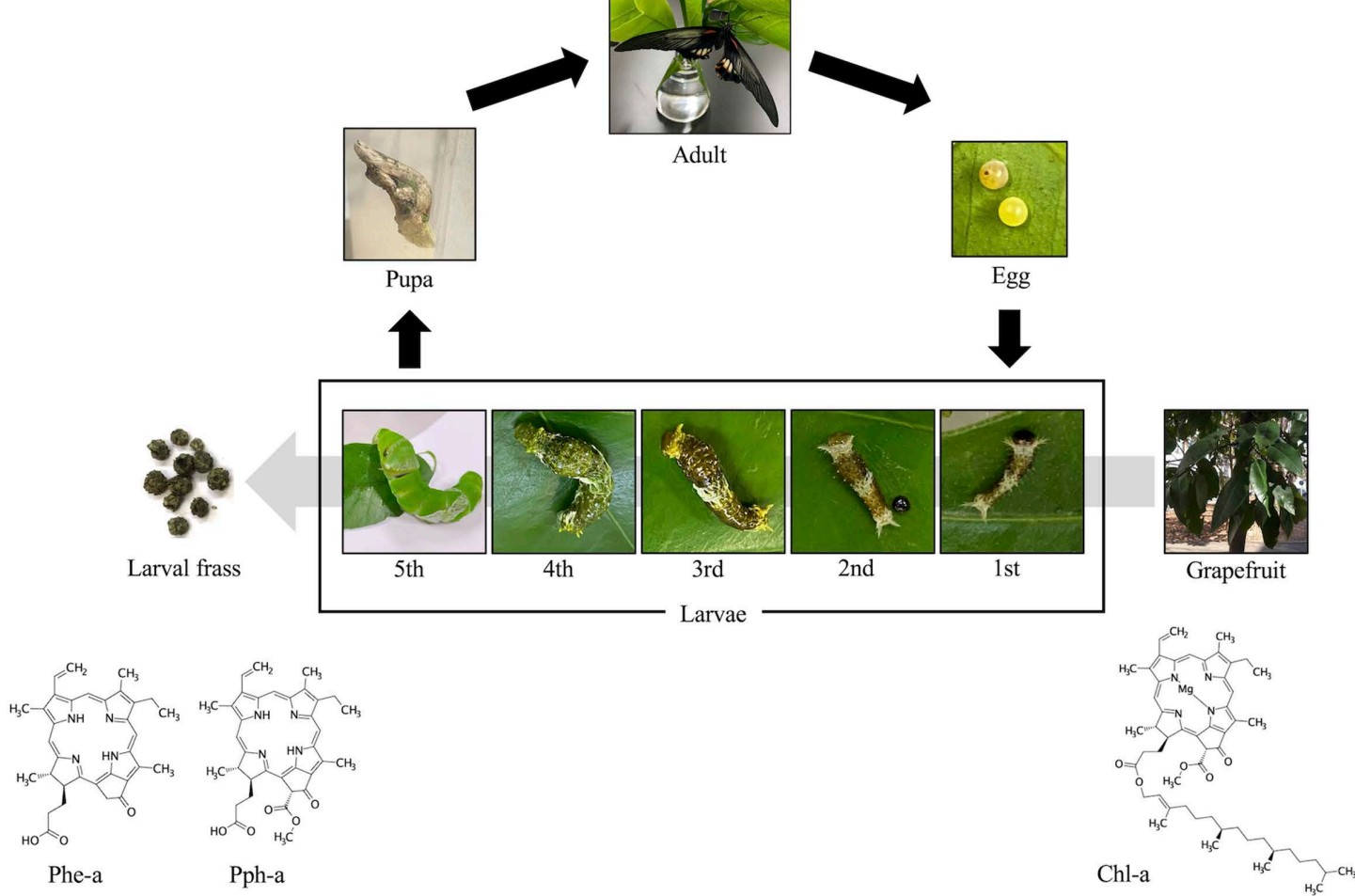

**Fig 7. Life cycle of *P. memnon* and alteration of chl-a to Phe-a and Pph-a via metabolism by *P. memnon* larvae.**

## Supporting information

**S1 Fig. The integrated $^1$H-NMR spectrum of Fr. 7 and Phe-a standard with the chemical structure.**
(TIFF)

**S2 Fig. The integrated $^1$H-NMR spectrum of Fr. 10 and Pph-a standard with the chemical structure.**
(TIFF)

**S3 Fig. The $^{13}$C-NMR spectrum of Fr. 10 and Pph-a standard with the chemical structure.**
(TIFF)

**S4 Fig. *CYP*s expression in the larval midgut and fat bodies.** (A-C) The expression levels of *CYP6B2* (A), *CYP6B4* (B), and *CYP6B5* (C) were analyzed by RT–qPCR (MG; midgut, FB; fat bodies). Relative expression levels mean relative quantification (RQ). Bars show RQ minimum and RQ maximum.
(TIFF)

**S5 Fig. Effects of the hexane and the methanol extracts on MIA PaCa2 cells.** (A-B) Viability of MIA PaCa2 cells relative to that of the control treatment with 0.1% (v/v) DMSO after treatment with Hex. extract (A) and the Met. extract (B) from larval frass and *C. paradisi* leaves (plant), as determined via the WST-8 assay. The error bars represent the means ± SDs from three biological replicates. (C-D) Cell morphological observations after treatment with Hex. extract (C) and the Met. extract (D) from the larval frass and *C. paradisi* leaves at 50 µg/mL. Scale bars = 100 µm.
(TIFF)

**S6 Fig. Effects of fractions from the chloroform extract from larval frass on MIA PaCa2 cells.** (A) Viability of MIA PaCa2 cells relative to that of the control treatment with 0.2% (v/v) DMSO after treatment with Chl. Fr., E.A. Fr., and Met. Fr. (B) Cell morphological observations after treatment with Chl. Fr., E.A. Fr., and Met. Fr. at 50 µg/mL. (C) Cell viability relative to that of the control treatment with 0.1% (v/v) DMSO after treatment with Fr. 1 to Fr. 3. (D) Cell morphology observation after treatment with Fr. 1 to Fr. 3 at 50 µg/mL. The cell viability was measured via a WST-8 assay. The error bars represent the means ± SDs from three biological replicates. Significant differences from the control were analyzed with Student's t test, *P < 0.05, ***P < 0.001. Scale bars = 100 µm.
(TIFF)

**S7 Fig. Peaks of fractions separated from Fr. 1 by HPLC.** Peaks were detected by 256 nm and 365 nm.
(TIFF)

**S8 Fig. MS analysis by MALDI-Spiral TOF.** (A) Precursor ion spectrum of Fr. 7 was *m/z* 592.3. (B) Precursor ion spectrum of Fr. 10 was *m/z* 534.3.
(TIFF)

**S9 Fig. MS analysis of Fr. 7, Fr. 10, and the standards by LC-ESI-MS.** (A) Comparison of Fr. 7 and the Phe-a standard. (B) Comparison of Fr. 10 and the Pph-a standard. Peaks were detected at 366 nm.
(TIFF)

**S1 Table. Primers used for RT–qPCR.**
(XLSX)

**S2 Table. The conditions of NMR spectroscopy.**
(XLSX)

**S3 Table. DEGs between the midgut and fat bodies in the *P. memnon* larvae analyzed by edgeR.**
(XLSX)

**S4 Table. Transcripts in the *P. memnon* larval midgut and fat bodies annotated with *P. xuthus* reference transcripts by BLAST.**
(XLSX)

**S5 Table. Comparison of ¹H-NMR chemical shifts among Fr. 7, Phe-a (standard), and Phe-a (literature) [40].**
(XLSX)

**S1 File. Primary NMR data files.**
(ZIP)

## Acknowledgments

We thank members of the Smart-Core-Facility Promotion Organization of Tokyo University of Agriculture and Technology for their technical assistance with MALDI-Spiral TOF-TOF and LC-ESI-MS analyses.

## Author contributions

**Conceptualization:** Miho Nakano, Yoshikazu Kitano, Hiroko Tabunoki.

**Data curation:** Miho Nakano, Takuma Sakamoto, Hidemasa Bono.

**Funding acquisition:** Miho Nakano, Takuma Sakamoto, Hidemasa Bono, Hiroko Tabunoki.

**Investigation:** Miho Nakano, Takuma Sakamoto.

**Methodology:** Miho Nakano, Takuma Sakamoto, Yoshiyuki Itoh, Yoshikazu Kitano, Kaori Tsukakoshi, Hidemasa Bono, Hiroko Tabunoki.

**Project administration:** Hiroko Tabunoki.

**Resources:** Miho Nakano, Takuma Sakamoto, Yoshiyuki Itoh, Yoshikazu Kitano, Kaori Tsukakoshi, Hiroko Tabunoki.

**Supervision:** Hiroko Tabunoki.

**Validation:** Takuma Sakamoto.

**Visualization:** Takuma Sakamoto, Yoshiyuki Itoh, Kaori Tsukakoshi.

**Writing – original draft:** Miho Nakano.

**Writing – review & editing:** Takuma Sakamoto, Yoshiyuki Itoh, Yoshikazu Kitano, Kaori Tsukakoshi, Hidemasa Bono, Hiroko Tabunoki.

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
