## [Decision Letter · Decision Letter 0]

PONE-D-25-12079The metabolic ability of swallowtails results in the production of bioactive substances from plant componentsPLOS ONE

Dear Dr. Tabunoki,

Thank you for submitting your manuscript to PLOS ONE. After careful consideration, we feel that it has merit but does not fully meet PLOS ONE’s publication criteria as it currently stands. Therefore, we invite you to submit a revised version of the manuscript that addresses the points raised during the review process.

We look forward to receiving your revised manuscript.

Kind regards,

Dave Mangindaan

Academic Editor

PLOS ONE

Journal Requirements:

2. We note that this submission includes NMR spectroscopy data. We would recommend that you include the following information in your methods section or as Supporting Information files:

1) The make/source of the NMR instrument used in your study, as well as the magnetic field strength. For each individual experiment, please also list: the nucleus being measured; the sample concentration; the solvent in which the sample is dissolved and if solvent signal suppression was used; the reference standard and the temperature.

2) A list of the chemical shifts for all compounds characterised by NMR spectroscopy, specifying, where relevant: the chemical shift (δ), the multiplicity and the coupling constants (in Hz), for the appropriate nuclei used for assignment.

3)The full integrated NMR spectrum, clearly labelled with the compound name and chemical structure.

We also strongly encourage authors to provide primary NMR data files, in particular for new compounds which have not been characterised in the existing literature. Authors should provide the acquisition data, FID files and processing parameters for each experiment, clearly labelled with the compound name and identifier, as well as a structure file for each provided dataset. See our list of recommended repositories here: https://journals.plos.org/plosone/s/recommended-repositories

Reviewers' comments:

Reviewer's Responses to Questions

**Comments to the Author**

1. Is the manuscript technically sound, and do the data support the conclusions?

Reviewer #1: Yes

Reviewer #2: Yes

2. Has the statistical analysis been performed appropriately and rigorously? 

Reviewer #1: Yes

Reviewer #2: Yes

3. Have the authors made all data underlying the findings in their manuscript fully available?

Reviewer #1: No

Reviewer #2: Yes

4. Is the manuscript presented in an intelligible fashion and written in standard English?

Reviewer #1: Yes

Reviewer #2: Yes

5. Review Comments to the Author

Reviewer #1: This study explores how Papilio memnon larvae metabolize Citrus × paradisi components, converting chlorophyll-a into pheophorbide-a (Phe-a) and pyropheophorbide-a (Pph-a). Using RNA-seq and various chemical separation techniques (TLC, HPLC, MS/MS, NMR), the authors analyze enzyme expression and identify metabolites. They also test the bioactivity of larval frass extracts on pancreatic cancer cells (MIA PaCa2) and Aβ1-42 aggregation, hinting at potential therapeutic applications. While the approach is novel, it still need more experimental support.

Major:

1. The authors hypothesize that the CYP6B family is involved in chlorophyll metabolism, but this has not been directly verified through gene knockout or in vitro enzyme activity assays. It is recommended to supplement the study with CYP6B gene silencing experiments to measure changes in Phe-a/Pph-a levels in larval frass or conduct recombinant protein enzyme activity assays to clarify its specific catalytic role in chlorophyll metabolism.

2. The anticancer effect of Pph-a is currently demonstrated only through in vitro cell experiments. It would be beneficial to include an in vivo mouse tumor model to extend the findings to physiological mechanisms.

3. The degradation of chlorophyll into Phe-a/Pph-a requires multiple enzymatic steps (e.g., chlorophyllase, Mg-dechelatase), but the study does not examine the expression or activity of related enzymes.

4. Quantifying the correlation between chlorophyll content in leaves and Phe-a/Pph-a levels in larval frass would provide stronger evidence for metabolic transformation.

5. The chemical stability of Phe-a/Pph-a in the frass extract should be assessed under different conditions, such as light exposure, temperature, and storage time.

Minor:

1. Several gene names in the manuscript are not italicized; please follow standard formatting conventions.

2. The descriptions of thin-layer chromatography (TLC) and column chromatography conditions are somewhat brief. It is suggested to provide more detailed experimental parameters and solvent system information.

3. The discussion section could be expanded to include comparative literature on chlorophyll metabolism and biotransformation in other insects to enhance the study’s broader context.

Reviewer #2: The authors have conducted the investigations and identified the production of pheophorbide-a and pyrophephorbide-a from chlorophyll via the metabolic functions of Papilio memnon larvae.

The work was done logically.

The experimental methods and approaches are routine.

The identified compounds were not new.

The research was straightforward but lacks novelty and significance, and the impact is not high.

The authors claim "Our findings may contribute to the development of a bioprocess for the

production of pheophorbide-a and pyrophephorbide-a from chlorophyll via the metabolic functions of P. memnon larvae", but it lacks convincing points on how to develop a bioprocess for the production, rather it seems not feasible and not economical to this reviewer.

6. PLOS authors have the option to publish the peer review history of their article (what does this mean? ). If published, this will include your full peer review and any attached files.

**Do you want your identity to be public for this peer review?** For information about this choice, including consent withdrawal, please see our Privacy Policy .

Reviewer #1: **Yes: ** Chen Ling

Reviewer #2: No

---

## [Author Response · Author response to Decision Letter 1]

7 Jun 2025

June 7, 2025

To Reviewer 1

Comment from Reviewer #1: This study explores how Papilio memnon larvae metabolize Citrus × paradisi components, converting chlorophyll-a into pheophorbide-a (Phe-a) and pyropheophorbide-a (Pph-a). Using RNA-seq and various chemical separation techniques (TLC, HPLC, MS/MS, NMR), the authors analyze enzyme expression and identify metabolites. They also test the bioactivity of larval frass extracts on pancreatic cancer cells (MIA PaCa2) and Aβ1-42 aggregation, hinting at potential therapeutic applications. While the approach is novel, it still need more experimental support.

Answer to comment: Thank you for your comment. We believe that our findings contribute to understanding the production process of pheophorbide-a and pyrophephorbide-a from chlorophyll, mediated by the metabolic capacity of Papilio memnon larvae. The metabolic enzymes in P. memnon larvae likely facilitate the conversion of host plant components into bioactive compounds. Because herbivorous insect frass can serve as a novel resource for identifying seed compounds, using these metabolic enzymes may enable the future production of bioactive compounds.

Many CYPs used in bioprocessing require a reaction temperature of 37°C. However, insect-derived enzymes function close to room temperature, making Papilio CYPs useful for developing bioprocess techniques without special temperature conditions. This approach could support global decarbonization efforts. As insect-derived metabolites remain underexplored, our findings may improve understanding of other insect bioactive compounds and CYPs. In the revised manuscript, we added new data to support our insights and altered text based on your suggestions and comments.

Suggestion 1: The authors hypothesize that the CYP6B family is involved in chlorophyll metabolism, but this has not been directly verified through gene knockout or in vitro enzyme activity assays. It is recommended to supplement the study with CYP6B gene silencing experiments to measure changes in Phe-a/Pph-a levels in larval frass or conduct recombinant protein enzyme activity assays to clarify its specific catalytic role in chlorophyll metabolism.

Answer 1: Thank you for highlighting this issue. We estimate that components from C. paradisi leaves may be metabolized by certain CYPs in the midgut and fat bodies of P. memnon larvae and excreted into frass, based on transcriptome data. Therefore, we do not claim that a specific CYP, such as CYPB6, is involved in this metabolic process described in the manuscript. However, expression levels of CYP6B2 (MSTRG.2586.1), CYP6B4 (MSTRG.9993.1), and CYP6B5 (MSTRG.8684.1) were upregulated in the larval midgut compared with the fat bodies, as shown through real-time quantitative PCR (RT-qPCR; S4 Fig and S1 Table). Additionally, our previous study (Nakano M. et al., Insects 2025) showed fluctuating CYP6B expression in the midgut and fat bodies of larvae fed various host plants. Thus, we cannot exclude the possibility that the CYP6B family contributes to aspects of chlorophyll metabolism. However, as you mentioned, future studies should investigate silencing or knockdown of CYP6Bs in P. memnon larvae. Therefore, we have revised the wording in the relevant sections and added a note on potential loss-of-function experiments (lines 99–101, 349–355, 570–572, 778–782, and 812–813; added references: 722–734).

Suggestion 2: The anticancer effect of Pph-a is currently demonstrated only through in vitro cell experiments. It would be beneficial to include an in vivo mouse tumor model to extend the findings to physiological mechanisms.

Answer 2: Thank you for this valuable suggestion. Our study is the first to report that Pph-a suppresses MIA PaCa2 cell viability and inhibits amyloid-beta aggregation. Pph-a derivatives have been reported to prevent tumor growth in A549 xenograft mouse models (Gao YH et al., 2020, 187, 11959) and reduce tumor size in MDA-MB-231 xenograft mouse models (Wang F et al., 2022, 203, 110328). Therefore, it is reasonable to expect that Pph-a may also affect the MIA PaCa2 xenograft mouse model. However, before conducting in vivo experiments, we must address several issues, such as whether the compound can be effectively absorbed from the intestine. Thus, we consider mouse experiments as the next step. Future studies should explore the physiological mechanisms in mouse models. The current study examined whether plant component changes from P. memnon metabolism affect biological activity. The findings suggest that herbivorous insects may become valuable medicinal resources. Accordingly, we revised the relevant text as follows:

“In this study, we investigated metabolic capacity by analyzing the final metabolites in larval frass and transcript levels of metabolic enzymes in the larval midgut and fat bodies of P. memnon larvae fed C. paradisi leaves” (lines 99–101).

“These results indicate that C. paradisi leaf components are primarily metabolized in the midgut of P. memnon larvae and excreted into larval frass” (lines 354–355).

“This study newly revealed that Phe-a and Pph-a inhibit Aβ1-42 aggregation. Elucidating their physiological mechanism using various mouse models is expected to clarify the effects of Phe-a and Pph-a.” (lines 570–572).

Suggestion 3: The degradation of chlorophyll into Phe-a/Pph-a requires multiple enzymatic steps (e.g., chlorophyllase, Mg-dechelatase), but the study does not examine the expression or activity of related enzymes.

Answer 3: We agree that, as you suggested, the degradation of chlorophyll into Phe-a/Pph-a involves several enzymatic steps. As previously mentioned, this study investigated whether plant component changes via P. memnon metabolism influence biological activity. Our transcriptome data (Table 1) show that several transcripts, including CYP6B2, CYP6B4, and CYP6B5, fluctuated in the midgut and fat bodies. Several studies, including our previous work, have reported that CYP6B may be involved in plant-derived component metabolism. We validated these CYP6Bs via RT-qPCR, finding that they were upregulated in the midgut of P. memnon larvae fed C. paradisi leaves (S4 Fig). Chlorophyll can be degraded into chlorophyllide by chlorophyllase or into chlorophyllin under alkaline conditions, such as those in the insect midgut. Chlorophyllide is known to bind red fluorescent protein (RFP), chlorophyll-binding protein (CHBP), and P252 protein in Bombyx mori (Oetama VSP et al., 2020, 76, 1–9). However, chlorophyllase has not been identified in insects, and it remains unclear whether RFP, CHBP, and P252 metabolize chlorophyllide into Pph-a and Phe-a. Moreover, chlorophyll metabolism in insects varies in efficiency across species (Badgaa A et al., 2014, 40, 1232–1240). Therefore, further research is needed to clarify the enzymatic steps in chlorophyll degradation in P. memnon larvae. We added these future prospects to the Discussion section (lines 529–539).

Suggestion 4: Quantifying the correlation between chlorophyll content in leaves and Phe-a/Pph-a levels in larval frass would provide stronger evidence for metabolic transformation.

Answer 4: Thank you for your suggestion. Given that the chloroform extract of frass significantly reduced MIA PaCa2 cell viability (Fig. 2A), we inferred that Pph-a content is higher in frass than in leaves. Furthermore, it has been reported that Phe-a and Pph-a levels in frass were approximately three- and five-fold higher, respectively, compared with chlorophyll content in host plant material in Spodoptera littoralis larvae (Badgaa et al., 2014, 40, 1232–1240). Therefore, we incorporated this insight into the manuscript (lines 544–549).

Suggestion 5: The chemical stability of Phe-a/Pph-a in the frass extract should be assessed under different conditions, such as light exposure, temperature, and storage time.

Answer 5: Thank you for this suggestion. Commercial Phe-a/Pph-a products are stable for four years at −20℃, according to their MSDS. The MSDS does not mention the chemical stability of Phe-a/Pph- under standard experimental conditions. We believe that Phe-a/Pph-a in the frass extract remained stable for at least 10 months at −20℃ in the dark. We stored frass in a freezer at −20℃, protected from the light, for 9 months before extraction and structural analysis. Pph-a was detected using MS 1 month after extraction, and Phe-a was detected via NMR/MS 11 months after extraction. Extracted components were stored at −20℃ in the dark, and extractions were performed in a standard experimental room without special conditions. These findings suggest that Phe-a/Pph-a in the frass extract did not decompose before chemical structure determination. Furthermore, the Pph-a–containing fraction reduced cell viability by 70% more than one year after extraction. Taken together, these observations indicate that Phe-a/Pph-a in frass extract remain stable for approximately 10 months. Therefore, we concluded that the chemical stability of Phe-a/Pph-a, including in frass, did not significantly affect compound structure determination in our study.

Minor:

Suggestion 1: Several gene names in the manuscript are not italicized; please follow standard formatting conventions.

Answer 1: Thank you for indicating these errors. We have placed gene names in italics (lines 342 and 346–347).

Suggestion 2: The descriptions of thin-layer chromatography (TLC) and column chromatography conditions are somewhat brief. It is suggested to provide more detailed experimental parameters and solvent system information.

Answer 2: Thank you for these comments. We moved the TLC descriptions after the HPLC descriptions in the Materials and Methods section to clarify the explanation of fraction names in the TLC method (lines 249–256). Furthermore, we added detailed descriptions for TLC and column chromatography in the Materials and Methods (lines 213, 215, 216, 218, 223, 230-231, 250–251, and 253–254).

Suggestion 3: The discussion section could be expanded to include comparative literature on chlorophyll metabolism and biotransformation in other insects to enhance the study’s broader context.

Answer 3: Thank you for your suggestion. Accordingly, we added content on chlorophyll metabolism and biotransformation in B. mori to the Discussion section (lines 529–539).

To Reviewer 2

Comment from Reviewer #2: The research was straightforward but lacks novelty and significance, and the impact is not high. The authors claim "Our findings may contribute to the development of a bioprocess for the production of pheophorbide-a and pyrophephorbide-a from chlorophyll via the metabolic functions of P. memnon larvae", but it lacks convincing points on how to develop a bioprocess for the production, rather it seems not feasible and not economical to this reviewer.

Answer to the comment: Thank you for your comments. We believe that our study will help advance the exploration of novel seed compounds in future research. This is especially important as we face limitations in obtaining sufficient seed compounds from plant resources owing to the depletion of new plant materials. Frass from herbivorous insects has long been used as a source of natural medicine in several Asian countries. Insect frass contains plant-derived compounds with chemical structures that are altered through insect metabolism. As a result, components in insect frass may show enhanced biological activity compared with the metabolites originally ingested from plants. However, the metabolic processes that modify the biological activity of host plant metabolites have not been well analyzed. Therefore, we examined the metabolic ability of P. memnon larvae fed Citrus × paradisi (grapefruit) and identified two chlorophyll catabolites, namely pheophorbide-a and pyropheophorbide-a, in the larval frass extract. These compounds not only exhibit biological activity, such as suppressing MIA PaCa2 cell viability and inhibiting amyloid-beta aggregation, but some of their derivatives demonstrate strong anticancer effects. The metabolic enzymes in P. memnon larvae may support the production of bioactive compounds and pharmaceutical intermediates from host plant components. Therefore, herbivorous insect frass could serve as a new resource for identifying seed compounds, and leveraging their metabolic enzymes may facilitate the production of bioactive compounds in future research.

Furthermore, CYP6B1 and CYP6B3 from Papilio polyxenes have been successfully expressed using an insect cell system (Wen Z et al., 2006, 23, 2434–2443). In contrast to Bacillus megaterium CYPs, which require incubation at 37°C, Papilio CYPs function at 30°C. This lower temperature requirement makes Papilio CYPs beneficial for bioprocessing with reduced environmental impact. In future studies, we will aim to express CYPs involved in metabolizing chlorophyll in P. memnon larvae to generate Phe-a/Pph-a by reacting the expressed CYPs with chlorophyll in vitro. These details have been added to the Discussion section (lines 578–583).

---

## [Decision Letter · Decision Letter 1]

The metabolic ability of swallowtails results in the production of bioactive substances from plant components

PONE-D-25-12079R1

Dear Dr. Tabunoki,

We’re pleased to inform you that your manuscript has been judged scientifically suitable for publication and will be formally accepted for publication once it meets all outstanding technical requirements.

Kind regards,

Dave Mangindaan

Academic Editor

PLOS ONE

Additional Editor Comments (optional):

Reviewers' comments:

Reviewer's Responses to Questions

**Comments to the Author**

1. If the authors have adequately addressed your comments raised in a previous round of review and you feel that this manuscript is now acceptable for publication, you may indicate that here to bypass the “Comments to the Author” section, enter your conflict of interest statement in the “Confidential to Editor” section, and submit your "Accept" recommendation.

Reviewer #1: All comments have been addressed

Reviewer #3: All comments have been addressed

2. Is the manuscript technically sound, and do the data support the conclusions?

Reviewer #1: Yes

Reviewer #3: Yes

3. Has the statistical analysis been performed appropriately and rigorously? 

Reviewer #1: Yes

Reviewer #3: Yes

4. Have the authors made all data underlying the findings in their manuscript fully available?

Reviewer #1: Yes

Reviewer #3: Yes

5. Is the manuscript presented in an intelligible fashion and written in standard English?

Reviewer #1: Yes

Reviewer #3: (No Response)

6. Review Comments to the Author

Reviewer #1: I have no other suggestions. All my previous questions have been addressed. No concerns about dual publication, research ethics, or publication ethics.

Reviewer #3: Review of PONE-D-25-12079R1

This is a well-written bioinformatics/computational biology manuscript. The characterization and the analysis are comprehensive. After seeing the serious revision made by the authors, this manuscript can be accepted now.

7. PLOS authors have the option to publish the peer review history of their article (what does this mean? ). If published, this will include your full peer review and any attached files.

**Do you want your identity to be public for this peer review?** For information about this choice, including consent withdrawal, please see our Privacy Policy .

Reviewer #1: **Yes: ** Chen Ling

Reviewer #3: No

---

## [Editor Report · Acceptance letter]

PONE-D-25-12079R1

PLOS ONE

Dear Dr. Tabunoki,

I'm pleased to inform you that your manuscript has been deemed suitable for publication in PLOS ONE. Congratulations! Your manuscript is now being handed over to our production team.

Kind regards,

on behalf of

Assoc. Prof. Dave Mangindaan

Academic Editor

PLOS ONE